# Dental Caries, Oral Health Behavior, and Living Conditions in 6–8-Year-Old Romanian School Children

**DOI:** 10.3390/children9060903

**Published:** 2022-06-16

**Authors:** Ramona Dumitrescu, Ruxandra Sava-Rosianu, Daniela Jumanca, Octavia Balean, Lia-Raluca Damian, Guglielmo Giuseppe Campus, Laurentiu Maricutoiu, Vlad Tiberiu Alexa, Ruxandra Sfeatcu, Constantin Daguci, Mariana Postolache, Atena Galuscan

**Affiliations:** 1Translational and Experimental Clinical Research Centre in Oral Health, Department of Preventive, Community Dentistry and Oral Health, University of Medicine and Pharmacy “Victor Babes”, 300040 Timisoara, Romania; dumitrescu.ramona@umft.ro (R.D.); jumanca.daniela@umft.ro (D.J.); balean.octavia@umft.ro (O.B.); damian.lia-raluca@umft.ro (L.-R.D.); vlad.alexa@umft.ro (V.T.A.); galuscan.atena@umft.ro (A.G.); 2Department of Restorative, Preventive and Pediatric Dentistry, University of Bern, Freiburgstrasse 7, 3012 Bern, Switzerland; guglielmo.campus@zmk.unibe.ch; 3Department of Psychology, West University of Timisoara, 300223 Timisoara, Romania; laurentiu.maricutoiu@e-uvt.ro; 4Oral Health and Community Dentistry Department, Faculty of Dental Medicine, UMP “Carol Davila”, 020021 Bucharest, Romania; 5Department of Oral Health, Faculty of Dentistry, University of Medicine and Pharmacy, 200585 Craiova, Romania; dagucicristi@yahoo.com; 6Department of Program Implementation and Coordination, Romanian Ministry of Health, 010024 Bucharest, Romania; mariana.postolache@ms.ro

**Keywords:** oral health, children, dental caries, behavior, prevalence, severity

## Abstract

Dental caries still have a high prevalence in Romania. The aim of this paper is to determine the prevalence of caries in children (aged 6 to 8 years) correlated with individual-level predictors and socio-economic variables. A stratified, randomized nationally representative sample was established, taking into consideration the total number of preschool children and based on administrative units and residence. Self-assessment was performed by means of the Oral Health Questionnaire for Children (WHO). Examinations were conducted by 10 standardized examiners, with International Caries Detection and Assessment System (ICDAS) caries codes higher than 3 considered as dentinal caries, missing teeth as *MT*, and restorations as *FT*. DMFT and SiC indexes were calculated accordingly. The dataset for each outcome variable was analyzed by the Hurdle approach analyzed. The gender distribution was similar (47.22% male and 52.78% female), with 42.65% residing in rural areas. The mean DMFT value for the sample was 4.89 and SiC index 9.83. A negative association could be seen between DMFT and the father’s level of education (β = −0.33, SE = 0.07, *p* < 0.01) as well as the mother’s education (β = −0.25, SE = 0.07, *p* < 0.01). In conclusion, caries prevalence is very high in Romania as compared to the World Health Organization (WHO) recommendation for this age group in correlation with socio-economic factors and oral health behavior.

## 1. Introduction

According to the WHO, major improvements can be seen regarding oral health, but worldwide there still exists a high burden of disease. The 2017 Global Burden of Disease states that tooth decay of primary and permanent teeth is the disease with the highest prevalence and the second highest incidence for oral conditions [1,2,3].

In recent decades, dental caries have remained the most important dental public health issue due to the high level of the affected population worldwide and especially in developing countries [4]. Dental caries are considered an oral disease and can be reversed if diagnosed in the early stages of enamel demineralization [5].

Although oral health does not endanger life, the impact on general health, on health care, and on the quality of life of the individual is very high, affecting sleep, intellectual activity, and self-esteem [6,7].

During the last decades, the epidemiology of dental caries has changed due to the implementation of oral health promotion programs and the increased use of fluoridated toothpastes, which are directly related to the reduction of caries and their complications [4,8,9]. This decreasing trend in the incidence of dental caries clearly supports the hypothesis that carious lesions can be monitored by monitoring risk factors. Furthermore, research shows that oral health risk factors are also common to other chronic diseases. Accordingly, oral health prevention programs have to be developed taking into consideration the general health promotion framework [4,8].

A strong relationship between socio-economic status and caries risk has been confirmed by several epidemiological studies. Socio-economic status (SES) has been described by Lee and Divaris (2014) [10] as being the most powerful “upstream” determinant of children’s poor oral health and plays an important role in health behavior, utilization of health services, and environmental exposure. Acces to oral care and diet quality are directly affected by deprivation; health behaviors are also influenced by social structure and environment, being considered a causal pathway for poor oral health in early years [11,12]. Children’s oral health behaviors, diet and feeding practices, and individual factors involved in cariogenesis could be effectively targeted by ensuring resources for primary school children even if these are limited. Good oral health behaviors consisting of regular tooth brushing, use of mouth washes, and flossing have been proven to be of importance in order to obtain good oral health and to reduce the burden of disease [11]. The role of sugar-containing foods in cariogenesis was priorly established [12]. However, there could be a significant variation in the importance of those factors in cariogenesis in comparison with other factors by taking into account other cultural practices or behaviours [13].

Dental caries still have a high prevalence in Romania. The Romanian dental care system is entirely private. A very small percentage of private practices have a collaboration with the National Insurance House, meaning that those practices can provide some free emergency and preventive treatments for school children up to the age of 18. Treatments are limited by a monthly budget and have to be temporized if this budget is exceeded. Distribution of dental offices is also uneven, mostly being present in urban areas, with very large rural areas having difficult access to care. There are no national prevention or education programs on oral health.

In Romania, no national data on caries prevalence or incidence is available and existing studies are focused on small samples selected locally. Furthermore, the data are not easily comparable due to different diagnostic criteria and the fact that non-standardized examiners were used. No comprehensive national epidemiological oral health surveys have been implemented in Romania. Results from two studies regarding caries prevalence in 12-year-olds have been published [14,15], suggesting a national mean DMFT for 12-year-olds of 4.1 in 1992 [14] and 2.8 in 2000 [15]. Thus, for this survey the International Caries Detection and Assessment System (ICDAS II) evaluation criteria were used in order to obtain results that are comparable to other similar studies worldwide. ICDAS II is a comprehensive evaluation system which allows detection of carious lesions starting from initial, non-cavitated stages on every tooth surface [16] by clinical examination. This enhances the quality of data collection by giving information both on the presence as well as the severity of dental caries. Early detection of caries is determinant for the management of such lesions. Non-cavitated lesions are nowadays considered reversible and there is a series of preventive procedures which can stop their evolution [17]. Minimal invasive therapies have been recommended to stop lesion progression and preserve hard dental tissues. A recent consensus meeting on dental caries recommended preserving the infected dental tissues without removing them and sealing the incipient lesions rather than conservative treatment [18]. For this reason, it is crucial to detect the extension of caries lesions and guide early prevention strategies [17].

The “Romanian Oral Health Survey” was designed as a cross-sectional survey for two age groups and conducted in 2019–2020. The aim of this paper was to describe the prevalence and severity of dental caries in a sample of 6-year-old children in Romania. A special interest was to correlate the prevalence and severity of dental caries with individual characteristics and socio-economic variables. The individual-level variables included gender, age, parents’ level of education, and dietary and preventative behaviors. The socio-economic variables examined were area, type of residence, and county development.

## 2. Materials and Methods

### 2.1. Study Design and Sample Selection

The “Romanian Oral Health Survey”, focused on 6- and 12-year-old children, was conducted in 2019–2020. This paper covers a sample of 6-year-olds. Ethical approval was sought from the Ministry of Health, Ministry of Education, and regional school inspectorates in Romania.

The first step consisted of sample selection and development of the questionnaire, which was validated for the targeted sample. Schools were selected according to the the total number of registered children in the 42 counties of Romania, and the sample size was estimated considering a ±3% sampling error at 95% confidence interval. Estimations suggested that a total of 1067 evaluations were required for each target population (i.e., 6 years and 12 years).

In order to create a nationally representative sample, the publicly available information found online for all 41 School Inspectorates and Bucharest, Romania’s capital, was collected and verified. For the 6-year-olds group, the number of children enrolled in pre-school was established. Publicaly available information obtained from all 4696 schools was evaluated, determining the distribution of the target community (Figure 1).

A national, stratified and randomized representative sample was created based on administrative units (counties) and types of residence (i.e., urban versus rural areas). The total number of pupils was determined for each county, and the resulting percentage of children was computed and was used to estimate the number of sampled children from each county. The total number of children was then divided based on rural or urban residence and the final sample size was obtained. To select schools from urban or rural areas, the randomization function (MS Excel) was used. A total of 49 schools were selected accordingly.

Considering the characteristics of the age groups, a Romanian version of the *Oral Health Questionnaire for Children* developed by the World Health Organization [19] was used to collect data. The adaptation process involved two independent translators that provided Romanian versions of the items. These versions were compared and the few differences in the translations were discussed in a meeting. To test the readability of the questionnaire in Romanian, the opinions of two psychologists were asked (educational psychology and developmental psychology) [19]. The questionnaires and informed consent were distributed to the children before the time of the clinical examination and were collected on the day they presented themselves for examination.

### 2.2. Clinical Examination

Of the subjects, 21 were examined during the standardization procedure by 10 examiners, fully registered dentists with experience in general dentistry and special interest in cariology. The inter-examiner kappa coefficient was 0.74–0.86, and the intra-examiner coefficient was 0.81–0.92. For fillings, an excellent value of the kappa coefficient was obtained, while for the severity of the lesion it ranged from good to excellent.

The examination procedure took place in classrooms or other available rooms, using examination kits and special front LED flashlights, commonly used in dentistry. The teacher was always present during the examination of the children. Examiners collected the Oral Health Questionnaires and informed consent completed and signed by parents or caregivers on the day of the examination. In order to remove plaque or food debris cotton rolls were used.

A special chart was used to register the ICDAS codes for each tooth surface. The system uses seven codes to assess the severity of caries lesions, 0 meaning sound enamel, without any sign of demineralization; code 1—visible signs of demineralization only after air drying; code 2—visible signs of demineralization without air drying; 3—enamel breakdown; 4—dentinal shadow; 5—cavity into dentin affecting less than half of the tooth surface; and 6—cavity into dentin affecting more than half of the tooth surface. Because examination was performed in a school setting and not the dental office, air drying was not possible, so codes 1 and 2 had to be counted together and marked as A, as described in the ICDAS examination protocol. The completed questionnaire and the informed consent for each child were attached to the examination chart.

### 2.3. Statistical Analysis

Dentine caries were considered those with an ICDAS index greater than 3, MT was the abbreviation for teeth missing due to severe carious lesions, and FT was the notation for restored dental surfaces. Thus, by calculating these indices, it was possible to analyze the variables related to caries (the dentinal caries index) and those associated with treatment of the lesions (the MT and FT indices). Children in this age group have mixed dentition. Therefore, they have both permanent and temporary teeth on the dental arches. The World Health Organization recommends taking surveys for this age group as well as the 12-year-olds, those being cut-off points for mixed dentition. Permanent teeth start erupting before the age of 6 and are fully erupted by 13 years of age.

The DMFT index and Significant Caries index (SiC) index were calculated accordingly. The DMFT index is used in epidemiology and represents the sum of decayed, missing, and filled teeth. Missing teeth are considered teeth that have been extracted due to complications of carious lesions; restored teeth are also considered as teeth treated by conservative treatment methods of irreversible caries lesions (fillings). Permanent teeth are scored with uppercase letters, temporary teeth with lowercase. For epidemiological purposes, the Significant Caries index was introduced in 2002 to express the highest values of the DMFT index. It represents the mean DMFT value of the highest one-third of scores. A multilevel approach was used to analyze the data because children were sampled from 49 schools. Each outcome variable was analyzed by the Hurdle approach. This means that for each variable, two sets of analyses were needed. The Bernouli estimation within a multilevel logistic regression (multilevel binary model) predicts prevalence for each outcome, meaning the presence or absence of each outcome for the entire sample. The multilevel analysis (Poisson) takes into consideration only the non-zero counts. To determine the level of dentinal caries, the Poisson multilevel overdispersed model was used. To analyze the restoration index (FT), a classical Poisson model was used. Each predictor was centered around a median value beacause it makes the interpretation of the analysis easier. Relationships between the three outputs and every predictor represented the point on which the analysis focused. Two regression analyzes were performed for each predictor: one multilevel logistic regression, the other one, a multilevel Poisson regression. The analyses are the same as zero-order correlations and determined the nested nature of results. Analyses were performed using HLM version 7.02, Scientific Software International [20].

Being the first Romanian national survey, this study looked at relationships between demographic parameters and those of oral health so that no adjustment for potential confounders was made. Adjustment should have been reported for partial correlations between oral health habits and parameters. Oral health behaviors were mostly not significantly correlated to demographic or socio-economic variables; this is the reason why controlling later variables would have had a small impact on the aforementioned parameters.

## 3. Results

A sample of 809 children having a mean age of 6.48 years were examined. The gender distribution was even over the sample (47.22% male and 52.78% female). A total of 42.65% of the children lived in rural communities and 57.35% in urban communities, with most in small communities. High school education was most common among the parents. (Table 1).

A percentage of 14.76% (106 children) had a DMFT index of 0. Mean DMFT of the sample was 4.89 and SiC index of the whole sample was 9.83.

A negative association could be seen between the DMFT and the parents’ level of education, being predicted by both the father’s education (β = −0.33, SE = 0.07, *p* < 0.01) and the mother’s educational level (β = −0.25, SE = 0.07, *p* < 0.01) (Table 2). More information regarding eating behaviors and caries prevalence can be found in the Appendix A.

The Poisson analysis described in Table 3 showed significant associations between DMFT and fruit consumption (β = 0.04, SE = 0.01, *p* < 0.05), fizzy drinks (β = 0.04, SE = 0.01, *p* < 0.01), milk (β = 0.05, SE = 0.01, *p* < 0.01), and tea (β = 0.05, SE = 0.01, *p* < 0.01). School-level predictors were negatively correlated to the dmft index (β = −0.54, SE = 0.25, *p* < 0.01).

## 4. Discussion

This study evaluated caries prevalence within a specific age group using ICDAS II criteria. Given the design of the study, examinations were performed in a classroom, not in the dental office. ICDAS II is a very comprehensive evaluation which allows complex staging of caries lesions and accounts for their severity. It evaluates caries severity as well as the activity of the lesions, starting from the point of active enamel demineralization [21]. Radiographs are commonly used in the dental office as a complementary assessment, mainly to determine pulpal involvement (lesions in proximity of, or affecting, dental pulp). This represents a complication of caries lesions or other traumas and it was not the purpose of this survey to investigate it. For this reason, visual examination was the most suitable method.

The DMFT value of 4.89 in 6-year-olds in Romania shows very high prevalence of dental caries for this age group, taking into consideration the WHO recommendation. This value is comparable to other countries in the region, Hungary having a DMFT of 4.5 [15], Croatia of 4.14 [15], Bosnia–Herzegovina of 6.7 [22], and Greece having the lowest DMFT index of 1.77 [23].

Looking at every category which sums up the DMFT index, the relationship between the incidence of missing teeth (the MT component) and preventative oral health–related behavior is influenced by the individual characteristics of the child (gender, age, parents’ education). Considering parents’ education, it can be deduced that the lack of education of one of the parents can determine the absence of preventative behaviors related to the oral health of 6–9-year-old children.

These results are consistent to another study in Belgium, where caries experience, expressed through DMFt and DMFs indexes, proved higher in groups that are underprivileged and family social context influences oral hygiene (plaque index) and the level of care [24]. Many international studies show evidence of the importance of social inequalities in oral health. Low socio-economic status has been shown to increase the risk for caries, especially in developed countries [25]. Low-income family status can majorly impact children’s oral health status in later years also. Studies done by Poulton et al. [26] concluded that increased levels of carious lesions and periodontal disease in adults are related to low socio-economic status in childhood, even after the adjustments for SES in adulthood. Listl et al. [27] agreed with these results, showing that financial problems in childhood have long-term adverse effects on oral health in adulthood. According to the obtained results, there are significant differences according to the county development index for the children in this age group.

Freire de Castilho [28] revealed, in a systematic review, that oral health habits of parents affect children’s oral health.

Demographic characteristics of the respondents investigated both mother’s and father’s levels of education. These predictors are of importance for the absence of lesions and the presence of restorations.

The probability of caries-free children is increased by a higher level of the parents’ education. Similar results were obtained for the presence of restoration. Children coming from families with highly educated parents have a higher chance of getting restorative treatment. Thus, the educational level of the parents could predict the presence of caries in children and, at the same time, can be considered as a useful predictor to determine the number of carious lesions. The more caries children have, the lower the level of the parents’ education is.

As previously reported, a high level of the father’s education strongly influences the socio-demographic pattern of caries, and the severity of caries could be affected by this parameter [29,30]. The fact that highly educated parents could influence good oral health habits for their children is an interesting speculation [31].

Dietary behavior seems to play a significant part in the appearance of dental caries, which are determined by excessive sugar consumption. From the obtained results, we can observe that there are statistically significant correlations between the presence of dental caries and consumption of sweetened milk, tea and cocoa, and fizzy drinks. Another important factor is represented by the region of residence, taking into account the geographical area and socio-economic development. These differences, in terms of health, affect society with deep consequences concerning the social, economic and political environment. Children who are likely to suffer from chronic illnesses in adulthood and perpetuate poverty and poor health over the generations are those children who currently live in poor families or housing, have an unhealthy diet, and lack access to education in their early childhood [32,33].

For this age group, gender was not significantly correlated with any of the components of the DMFT index, whereas there was an association between gender and number of restored teeth for the 12 year olds [20]. This could be explained by cultural factors, with girls getting more attention in contrast to boys, where appearance is concerned, when they grow up. Another study in Scotland discussed the fear factor being higher in boys than in girls where seeking dental treatment is concerned [34].

The strength of this study is represented by the complex sample selection process, the statistical analysis, and the standardization of examiners. The standardization process of the examiners as well as the complex way of selecting the sample and analyzing the data support the validity of the results and findings obtained.

Being the first national study to evaluate both caries prevalence as well as caries risk factors and behaviors is another strong point. The STEPs survey method was described by the WHO to insure the evaluation of trends at the national level and to allow for the possibility of comparing those to other countries. By this means, the collection of data can be done on regular and continuing basis. Regarding the limitations of the study, one of them could be the interpretation of the results. Due to the design of the study, the results obtained could be biased. The explanation for this is that cross-sectional studies measure both cause and effect in the same time interval; thus, there is a problem of time-related ambiguity and the inability to determine causal relationships. Additionally, one of the limits of the survey may be the methodological difficulties related to the organization of data collection in the 42 counties and the level of non-registration in the participation rate. In addition, it should be borne in mind that the socio-economic data obtained from the questionnaires may not be sufficiently reliable.

## 5. Conclusions

Caries prevalence is very high in Romania as compared to the WHO recommendation for this age group. The prevalence of caries in children from Romania is influenced by parameters such as the level of the parents’ education, lack of a healthy diet, and the county development index.

## Figures and Tables

**Figure 1 children-09-00903-f001:**
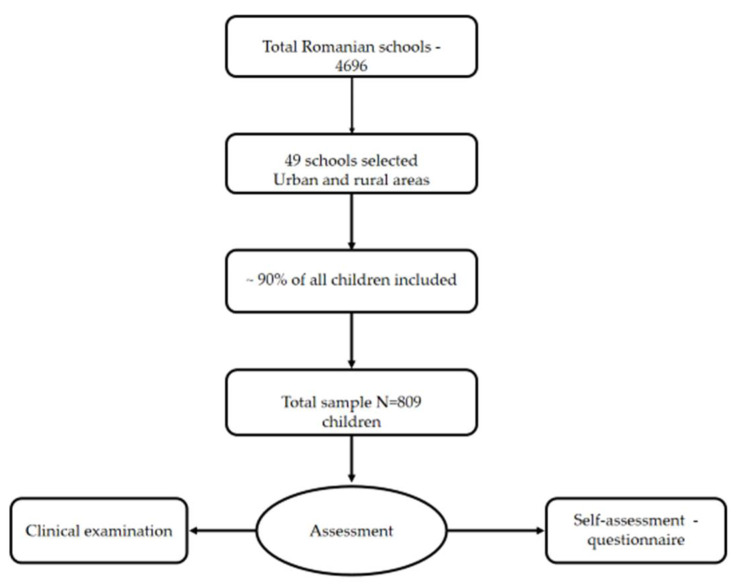
Study design flowchart.

**Table 1 children-09-00903-t001:** Descriptive statitstics of the sample.

Variables	N (%)	Variables	N (%)
**Gender**		**Father’s education**	
Male	382 (47.22%)	No school	19 (2.35%)
Female	427 (52.78%)	Primary school (4th grade)	30 (3.71%)
**Age**		Gymnasium (8th grade)	109 (13.47%)
6 y.o.	438 (54.14%)	Secondary school (10th grade)	155 (19.16%)
7 y.o.	342 (42.27%)	High school (12th grade)	226 (27.94%)
8 y.o.	22 (2.72%)	Post-secondary school	24 (2.97%)
Did not declare	7 (0.87%)	University studies	203 (25.09%)
**Residence**		Do not know/Did not answer	28 (3.46%)
Rural area	345 (42.65%)	Do not have a father figure	15 (1.85%)
Urban area	464 (57.35%)	**Mother’s education**	
**City type**		No school	24 (2.97%)
Small town	520 (64.28%)	Primary school (4th grade)	25 (3.09%)
Middle-sized city	56 (6.92%)	Gymnasium (8th grade)	107 (13.23%)
Big city	233 (28.80%)	Secondary school (10th grade)	124 (15.33%)
		High school (12th grade)	204 (25.22%)
		Post-secondary school	49 (6.06%)
		University studies	256 (31.64%)
		Do not know/Did not answer	18 (12.22%)
		Do not have a mother figure	2 (0.25%)

**Table 2 children-09-00903-t002:** Correlation of DMFT and its separate components with individual-level predictors and school-level predictors—Bernoulli prediction; d_3_t—dentinal caries; mt—missing teeth; rt-restored teeth; DMFT—sum of decayed, missing, and restored teeth; N = 809; * *p* < 0.05; ** *p* < 0.01.

	Occurrence vs. 0. Non-Ocurrence of Carious Lesions (Multilevel Logistic Regression Bernouli Estimations)			
	mt	d_3_t	rt	DMFT
**Individual-Level Variables**				
Gender (−1 boy, 1—girl)	−0.01 (0.20)	−0.01 (0.16)	−0.03 (0.17)	−0.10 (0.20)
Father’s level of education	−0.10 (0.06)	−0.31 ** (0.05)	0.08 (0.06)	−0.33 ** (0.06)
Mother’s level of education	−0.07 (0.07)	−0.29 ** (0.06)	0.22 ** (0.06)	−0.25 ** (0.07)
How often cleans teeth (0—“never”)	0.62 (0.09)	−0.23 ** (0.08)	0.14 (0.09)	−0.15 (0.10)
Fruit consumption (0—“never”)	−0.10 (0.11)	0.16 (0.09)	0.14 (0.10)	0.13 (0.12)
Pastry (0—“never”)	0.24 * (0.12)	0.05 (0.09)	−0.22 (0.10)	0.14 (0.12)
Fizzy drinks (0—“never”)	0.10 (0.09)	0.18 * (0.07)	−0.15 (0.08)	0.07 (0.09)
Honey (0—“never”)	0.12 (0.09)	0.02 (0.07)	−0.06 (0.08)	0.01 (0.09)
Chewing gum (0—“never”)	−0.01 (0.09)	0.22 * (0.08)	−0.14 (0.09)	0.17 (0.10)
Candies (0—“never”)	0.15 (0.09)	0.26 * (0.08)	−0.09 (0.08)	0.19 * (0.09)
Milk (0—“never”)	−0.03 (0.08)	0.16 * (0.07)	−0.09 (0.07)	0.12 (0.08)
Tea (0—“never”)	−0.01 (0.09)	0.21 * (0.08)	0.08 (0.09)	0.18 (0.10)
Cocoa (0—“never”)	−0.02 (0.10)	0.11 (0.08)	−0.01 (0.09)	0.−10 (0.10)
**School-level variables**				
Type of residence (−1 Rural, 2 Urban)	0.22 (0.35)	−0.57 (0.22)	10.0 * (0.35)	−0.54 ** (0.25)
Development index (County, centered on the mean)	−0.01 (0.01)	−0.01 (0.00)	0.02 (0.01)	−0.00 (0.01)

**Table 3 children-09-00903-t003:** Correlation of DMFT and its separate components with individual-level predictors and school-level predictors—Poisson correlation; d_3_t—dentinal caries; rt-restored teeth; DMFT—sum of decayed, missing, and restored teeth; N = 809; * *p* < 0.05; ** *p* < 0.01.

	Poisson Analysis—Regressions of Non-Zero Count		
	d_3_t	rt	DMFT
**Individual-level variables**			
Gender (−1 boy, 1—girl)	−0.16 ** (0.02)	0.14 (0.07)	−0.06 * (0.03)
Father’s level of education	−0.06 ** (0.00)	0.08 ** (0.02)	−0.04 ** (0.01)
Mother’s level of education	−0.05 ** (0.01)	0.14 ** (0.03)	−0.03 ** (0.01)
How often cleans teeth (0—“never”)	−0.03 * (0.01)	0.14 ** (0.04)	−0.02 (0.01)
Fruit consumption (0—“never”)	0.03 ** (0.01)	0.03 (0.05)	0.04 * (0.01)
Pastry (0—“never”)	0.02 * (0.01)	−0.05 (0.048)	0.02 (0.019)
Fizzy drinks (0—“never”)	0.033 ** (0.01)	−0.20 ** (0.04)	0.03 ** (0.01)
Honey (0—“never”)	0.02 * (0.01)	0.09 * (0.04)	0.03 * (0.01)
Chewing gum (0—“never”)	−0.03 ** (0.01)	−0.16 ** (0.04)	0.00 (0.00)
Candies (0—“never”)	−0.01 (0.01)	−0.07 (0.04)	0.03 (0.01)
Milk (0—“never”)	0.03 ** (0.01)	−0.06 (0.03)	0.05 ** (0.01)
Tea (0—“never”)	0.04 ** (0.01)	−0.00 (0.04)	0.05 ** (0.01)
Cocoa (0—“never”)	0.05 ** (0.01)	−0.09 * (0.04)	0.04 * (0.01)
**School-level variables**			
Typeof residence (−1 Rural, 2 Urban)	−0.18 (0.11)	−0.04 (0.12)	−00.04 (0.07)
Development index (County, centered on the mean)	−0.00 (0.00)	0.00 (0.00)	−0.00 (0.00)

## Data Availability

The data presented in this study are available on request from the corresponding author. The data are not publicly available in accordance with the consent provided by participants on the use of confidential data.

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
