# Peer review of "Dental Caries, Oral Health Behavior, and Living Conditions in 6–8-Year-Old Romanian School Children"

_children, 2022, doi:10.3390/children9060903_

Round 1
Reviewer 1 Report
Thank you for the opportunity to review this manuscript. Monitoring of population oral health, and especially child oral health is important and a core function of dental public health. The authors need to be congratulated for carrying out what seems to be the first national level child oral health survey.
Only some minor suggestions:
p1. line 1: change "temporary" to "primary"
p2. line 55: change "carious lesion" to "carious lesions"
p3. line 90: change to "6 and 12-year old children"
p3. line 91: change to "...covers the sample of 6-year olds"
Include a statement somewhere on ethics approval to carry out the study. Who or which committee/organization granted ethical approval? (This is important as it involved oral examinations carried out on young children).
What is the total number of 6-year olds in Romania (se readers get a sense of how representative the sample is)
Correct the word "Descriptive" in the title of Table 1. (The "e" was omitted)
Page 8. line 206 Change to "Looking at every component.."
Just add some information in the Introduction on the oral health care system for children in Romania. Do they have good access? Or access problems in some areas? Do they have to pay for services, or are there some subsidized dental services for children. (Affordability in general) Is it delivered by the public or private (or both) sectors?
Author Response
Dear reviewer,
Thank you very much for your suggestions and for the kind words regarding our study.
All the suggested corrections have been done in the main text as follows:
p1. line 1: changed "temporary" to "primary"
p2. line 55: changed "carious lesion" to "carious lesions"
p3. line 90: changed to "6 and 12-year old children"
p3. line 91: changed to "...covers the sample of 6-year olds"
Include a statement somewhere on ethics approval to carry out the study. Who or which committee/organization granted ethical approval? (This is important as it involved oral examinations carried out on young children).
R: A statement regarding ethical approval has been added to the Material and methods section, as follows: Ethical approval was obtained from the Romanian Ministry of Health, Romanian Ministry of Education and regional school inspectorates
What is the total number of 6-year olds in Romania (so readers get a sense of how representative the sample is)
R: The number of children enrolled in the first year of school (grade 0) for each Romanian school was established using public information available on the websites of all County School Inspectorates (41 counties plus the Romanian Capital). This process involved the evaluation of publicly available information from 4696 schools, which aided our under-standing of the territorial distribution of the target population. The sample was stratified and randomized. The stratification was performed on the administrative units (counties), and on locality type (i.e., urban versus rural localities). For each county, the total number of pupils was determined and expressed as a percentage share related to the total number of children. The percentage share was used to estimate how many children would have to be included in each county. The obtained number was then divided based on locality type (i.e., urban versus rural) and the final target number of evaluations was obtained.
Correct the word "Descriptive" in the title of Table 1. (The "e" was omitted)
Page 8. line 206 Change to "Looking at every component.."
R: corrections have been done in the text.
Just add some information in the Introduction on the oral health care system for children in Romania. Do they have good access? Or access problems in some areas? Do they have to pay for services, or are there some subsidized dental services for children. (Affordability in general) Is it delivered by the public or private (or both) sectors?
R: The following paragraph has been added to the introduction
“The Romanian dental care system is entirely private. A very small percentage of private practices have a collaboration with the National Insurance House, meaning that those practices can provide some free emergency and preventive treatments for school children up to the age of 18. Treatments are limited by a monthly budget and have to be temporized if this budget is exceeded. Distribution of dental offices is also uneven, mostly being present in urban areas, very large rural area having difficult access to care. There are no national prevention or education programmes on oral health.”
Reviewer 2 Report
I read the manuscript with great interest.
Abstract:
Concise and informative
Keywords: Use Mesh terms
Introduction:
The purpose of the study should be mentioned clearly
rephasing of the last paragraph in the introduction suggested.
Methods:
the sample size and distribution little confusing kindly rewrite this content
ICDAS code was used by the authors
this is very old now since the present study has been done already kindly explain the use of ICDAS
selection of the children and examination also a little confusing, kindly rewrite this content.
Results:
well written and clearly explained.
Figure 1 may not be useful.
demographic characteristics should be stated in a table.
The study design flowchart is recommended.
Discussion:
The discussion is very interesting.
The recent studies from Romania from my point of view are good to compare
Strengths and limitations give additional value to the study, authors are requested to make the last paragraph discuss strengths and limitations.
Reference:
recent studies are recommended.
Author Response
Dear reviewer,
Thank you for accepting to review our study and for your suggestion. We tried to respond to all your comments as follows
Keywords: Use Mesh terms
R: Mesh terms have been added to keywords: oral health; children; dental caries; behaviour; prevalence; severity.
Introduction:
The purpose of the study should be mentioned clearly rephrasing of the last paragraph in the introduction suggested.
R: The aim has been rephrased as follows: This paper aims to describe caries prevalence and severity in 6-year-old schoolchildren and correlate it individual level predictors such as gender, age, parents’ education, dietary and preventative behaviour and socio-economic variables, such as area and type of residence and county developmental index.
Methods:
the sample size and distribution little confusing kindly rewrite this content
R: the paragraph regarding sample size and selection has been rephrased as follows: Considering the characteristics of the age groups, the WHO Oral Health Questionnaire for Children was used to collect data. Regarding the readability of the Romanian version of the Oral Health Questionnaire for Children, the opinions of two experts were requested (i.e. PhD in educational psychology and developmental psychology). The questionnaires and informed consent were distributed to the children before the time of the clinical examination and were collected on the day they presented themselves for examination.
ICDAS code was used by the authors, this is very old now since the present study has been done already kindly explain the use of ICDAS selection of the children and examination also a little confusing, kindly rewrite this content.
The International Caries Detection and Assessment System (ICDAS) has been constantly developed over the last two decades and represents nowadays the basis of the International Caries Classification and Management System (ICCMS). It registers both prevalence and severity of caries lesions, allowing for a comprehensive evaluation and management plan of the lesions within the ICCMS 4D plan: Determine (patient level caries risk) – Detect (caries staging and activity) – Decide (Personalized care plan at patient and tooth level) – Do (appropriate prevention, intervention and control). The system uses seven codes to assess the severity of caries lesions, 0 meaning sound enamel, without any sign of demineralization, code 1 - visible signs of demineralization only after air drying, code 2 – visible signs of demineralization without air drying, 3 – enamel breakdown, 4 – dentinal shadow, 5 – cavity into dentin affecting less than half of the tooth surface and 6 – cavity into dentin affecting more than half of the tooth surface. Because examination was performed in a school setting and not the dental office, air drying was not possible so codes 1 and 2 had to be counted together and marked as A, as described in the ICDAS examination protocol.
Results:
well written and clearly explained. Figure 1 may not be useful. demographic characteristics should be stated in a table. The study design flowchart is recommended.
R: Figure 1 was removed and a graphical flowchart was added.
Discussion:
The discussion is very interesting. The recent studies from Romania from my point of view are good to compare
R: Up until now there have been no national oral health surveys in Romania. The last survey was performed by Professor Poul Eric Petersen in 1992 covering only six urban areas. The methodology of the study isn’t clearly available, therefor it is very difficult to compare it to our survey. The aim of the present survey was to gather baseline information on the oral health status of children in Romania, using the comprehensive STEPs approach described in the WHO guidelines for health surveys.
Strengths and limitations give additional value to the study, authors are requested to make the last paragraph discuss strengths and limitations.
R: the following paragraph regarding strengths and limitations has been added:
The strength of this study resides in the validity of the findings is supported by the complex process of the sample selection and data analysis as well as by the thorough calibration process. Another strong point is that the present study is the first one carried out in Romania that takes into account caries prevalence and also evaluates the role of the main caries risk factors and oral health-related behaviours, using the STEPS approach as suggested by the WHO guidelines. This ensures the possibility of evaluating national trends as well as comparing these to other countries. The STEPS approach advises the collection of data on a regular and continuing basis.
One of the limitations of the study resides in the interpretation of the results, which might be biased because of the study’s design, as, in general, a cross-sectional study measures cause and effect at the same point in time, therefore introducing the problem of temporal ambiguity and an inability to establish causal relationships. Among the limitations of this survey, it is possible to recognize some methodological difficulties in organizing and managing the data collection across the 42 counties, and not registering the participation rate can be considered as one of the limitations of this study. Additionally, it is necessary to take into account that the questionnaire data source may not be reliable enough, specifically in relation to socioeconomic data.
Round 2
Reviewer 2 Report
All the queries have been addressed.
No manuscript looks better in shape.
However, kindly resubmit the manuscript without track changes.
Highlight the changes in RED color.
The introduction should be improved and still the purpose is not clear.
The below references might be useful for discussion.
ignori C; CaCIA collaborative group, Uehara JLS, Romero VHD, Moro BLP, Braga MM, Mendes FM, Cenci MS. Comparison of two clinical approaches based on visual criteria for secondary caries assessments and treatment decisions in permanent posterior teeth. BMC Oral Health. 2022 Mar 18;22(1):77. doi: 10.1186/s12903-022-02112-6.
Dos Santos NM, Leal SC, Gouvea DB, Sarti CS, Toniolo J, Neves M, Rodrigues JA. Sealing of cavitated occlusal carious lesions in the dentine of deciduous molars: a two-year randomised controlled clinical trial. Clin Oral Investig. 2022 Jan;26(1):1017-1024. doi: 10.1007/s00784-021-04085-2.
Bhumireddy JR, Nirmala S V, Malineni SK, Nuvvula S. Diagnostic performance of the visual caries classification of International Caries Detection and Assessment System II versus conventional radiography for the detection of occlusal carious lesions in primary molars. SRM J Res Dent Sci 2019;10:117-21
Lin YT, Chou CC, Lin YJ. Caries experience between primary teeth at 3-5 years of age and future caries in the permanent first molars. J Dent Sci. 2021 Jul;16(3):899-904. doi: 10.1016/j.jds.2020.11.014.
Macey R, Walsh T, Riley P, Glenny AM, Worthington HV, O'Malley L, Clarkson JE, Ricketts D. Visual or visual-tactile examination to detect and inform the diagnosis of enamel caries. Cochrane Database Syst Rev. 2021 Jun 14;6(6):CD014546. doi: 10.1002/14651858.CD014546.
Author Response
Dear reviewer,
Thank you for your kind suggestions. The authors made the following changes to the original draft:
- The manuscript was submitted without track changes.
- The changes are now highlighted in RED colour.
- The introduction has been improved according to the recommendations and the purpose has been rephrased.
- The suggested references have been added in the introduction and discussion.
